# Compliance of Static Stretching and the Effect on Blood Pressure and Arteriosclerosis Index in Hypertensive Patients

**Etsuko Yamada [1], Sayuri Sakai [1], Mieko Uchiyama [1], Hansani M. Abeywickrama [1], Masanori Inoue [2], Kazuo Maeda [3], Yuko Kikuchi [1,4], Kentaro Omatsu [1,5] and Yu Koyama [1,\*]**

1   Department of Nursing, Niigata University Graduate School of Health Sciences, 2-746 Asahimachi, Niigata 951-8518, Japan; b17a306e@mail.cc.niigata-u.ac.jp (E.Y.); sakai@clg.niigata-u.ac.jp (S.S.); uchiyama@clg.niigata-u.ac.jp (M.U.); hansani@clg.niigata-u.ac.jp (H.M.A.); b20a301f@mail.cc.niigata-u.ac.jp (Y.K.); b21a301a@mail.cc.niigata-u.ac.jp (K.O.)
2   Inoue Internal Medicine Clinic, 5-3-5 Meikekaminoyama, Niigata 950-0945, Japan; inouenaika@able.ocn.ne.jp
3   Maeda Internal Medicine Clinic, 3160-4 Akebono, Niigata 951-8021, Japan; maeda-4560@mocha.ocn.ne.jp
4   Nursing Unit, Gosen Central Hospital, 489-1 Ohta, Gosen 959-1825, Japan
5   Division of Emergency Medical Sciences, Niigata University of Health and Welfare, 1398 Shimami-cho, Niigata 950-3198, Japan
\*   Correspondence: yukmy@clg.niigata-u.ac.jp; Tel./Fax: +81-25-227-2361

**Abstract:** Background: Treatment of high blood pressure is a combination of lifestyle changes and medications, and appropriateexercise therapy is recommended as one of the lifestyle-related changes. Recently, stretching, a low-intensity exercise, was reported to be antihypertensive and effective for improving arteriosclerosis, in addition to aerobic exercise. The present study investigated the short-term effects of continuous stretching and rest-induced rebound on vascular endothelial function in hypertensive patients. Methods: This study was conducted as a single-arm prospective interventional study including patients between 30 and 70 years of age undergoing treatment for hypertension from October 2019 until May 2021. The intervention consisted of six months of daily stretching, one month of rest, and another three months of stretching. We measured arteriosclerosis indices such as cardio ankle vascular index (CAVI), ankle brachial pressure index (ABI) and reactive hyperemia index (RHI), and flexibility at the baseline and one, three, six, seven, and ten months from the baseline. Results: We included a total of ten patients (three males and seven females) with an average age of 60.10 ± 6.05 years. The exercise rate for the entire period was 90% or more, and the anteflexion measurement value improved significantly before and after the intervention ($p < 0.001$). Blood pressure and CAVI/ABI were well controlled throughout the study period. RHI did not show any significant improvement during the initial six months, and only slightly improved by the third month ($p = 0.063$). Even after the rest phase and resumption of stretching, RHI remained stable. Conclusions: The compliance of the stretching program we used, evaluated by the exercise implementation rate for the entire period, was 90% or more; therefore, easy to perform and continue by hypertensive patients. However, we did not observe a significant positive effect on arteriosclerosis index or blood pressure in this study.

**Keywords:** hypertension; muscle stretching exercises; compliance

## 1. Introduction

Due to global population growth and aging, the number of hypertensive patients has doubled from approximately 650 million in 1990 to approximately 1.28 billion in 2019. Whereas the prevalence of hypertension is mainly declining in high-income countries, it is rising in low- and middle-income countries, and this trend is expected to continue [1]. Factors favoring the development of hypertension include obesity, an unhealthy diet, excessive dietary sodium, inadequate dietary potassium, inadequate physical activity, and alcohol consumption [2]. Therefore, hypertension is treated by combining lifestyle

modifications, such as salt reduction, maintenance of proper weight, alcohol restriction, smoking cessation, and exercise therapy, with drug treatment.

Increased physical activity not only lowers blood pressure [3], and an accompanying reduction of body weight and fat also improves insulin sensitivity, suppresses the development of type 2 diabetes, improves serum lipids, and maintains skeletal muscle mass. Additionally, there are reports that improved physical fitness is vital to the primary prevention of mental health disorders and dementia [4–6]. Therefore, appropriate exercise therapy is strongly recommended in hypertensive patients. However, the published reports indicate that the number of people exercising enough to prevent high blood pressure is low. According to the 2018 WHO Report [7], 81.0% of young people (ages 11–17) do not exercise for more than an hour a day. One in four adults (aged 18 and over), or 1.4 billion, are considered under-exercised. Even though the importance of performing and continuing exercise training for blood pressure control is widely recognized, only 15% of US adults have been found to meet exercise training/physical activity recommendations [8]. Because most aerobic, resistance, and combined aerobic and resistance training exercises are moderate intensity and time-consuming tasks [9], it can be quite challenging to continue these exercises as daily actions for working people. In addition, the studies that evaluated the continuity of exercises are scarce. In light of the above facts, we thought to adopt a low-intensity and less time-consuming stretching exercise protocol in this study.

While the antihypertensive effect of aerobic exercise has been established, it has also recently been reported that stretching, which is a low-intensity exercise, also produced an improvement in arteriosclerosis and an antihypertensive effect [10,11]. Daily muscle stretching for one month promoted endothelium-dependent vasodilation of skeletal muscle-resistant arterioles in aged rats [12]. It has also been reported that the pulse wave velocity decreases significantly until 30 min after stretching [13], and that it decreases only on the stretched side [14], which indicates an acute effect of stretching on the vascular endothelial function. According to the findings of these studies, stretching is expected to improve the arteriosclerosis index and antihypertensive effect, and it is highly possible that stretching can be recommended as exercise support for hypertensive patients. If stretching can improve vascular endothelial function, it will expand the exercise options for hypertensive patients in their daily life and clinical exercise therapy. However, such beneficial effects of stretching cannot be expected unless it can be easily performed and continued by hypertensive patients. Therefore, the purpose of this study was to investigate the compliance and continuity of stretching for relatively short-term hypertensive patients and evaluate its effect on vascular endothelial function.

## 2. Materials and Methods

### 2.1. Study Design and Participants

This study was conducted with a single-arm, prospective interventional study design at two internal medicine out-patient clinics in Niigata city, Japan, between October 2019 and May 2021, and was carried out with the approval of the Institutional Review Board of Niigata University (#2018-0025).

Participants, who were receiving treatment for hypertension, were included. The inclusion criteria for this study were age between 30 and 70 years and a diagnosis of essential hypertension. Exclusion criteria were secondary hypertension, organic heart disease, having quit smoking, and those who were judged inappropriate by doctors. Patients in the hospital were recruited for the study by posting posters in the facility and handing out leaflets. The patients who wished to participate were given both written and verbal information before providing their informed consent. Ten participants were included.

The intervention consisted of six months of daily stretching, followed by one month of rest, and then another three months of stretching. The researchers explained the content and precautions of the stretches and that they should be done daily from the next day before measuring blood pressure at home. The researchers measured the arteriosclerosis

index and flexibility in the subjects at the baseline and one, three, six, seven, and ten months from the baseline (Figure 1).

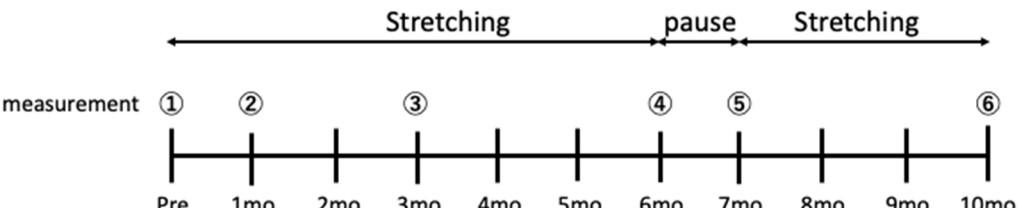

**Figure 1.** Study protocol. The stretching intervention consisted of six months of daily stretching, one month of rest, and another three months of daily stretching. The arteriosclerosis indices (RHI, CAVI, and ABI) and flexibility (forward-bending value) were measured at baseline and one, three, six, seven, and ten months from the baseline. Abbreviations: RHI, reactive hyperemia index; RH-PAT index: reactive hyperemia peripheral arterialtonometry index; CAVI, cardio ankle vascular index.

For the intervention in this study, we adopted the stretching protocol, which was originally recommended by the Japanese Ministry of Health, Labour, and Welfare to prevent locomotive syndrome in hypertensive patients. The stretching protocol is based on static stretching, and includes legs, trunk, shoulders, and arms. (https://www.mhlw. go.jp/www1/topics/kenko21_11/b2.html, accessed on 31 January 2022). The stretching program we used this time consists of 10 elements: calf, back of the thigh, front of the thigh, inside of thigh, waist, upper back, neck, shoulder, upper arm, and wrist. One series took about 10 min and was a low-intensity static stretch exercise that could be easily performed during breaks at home or work (Table 1).

**Table 1.** Stretching protocol.

| Stretching Site | Procedure |
| --- | --- |
| 1. Calf | While holding on to a chair, place the left leg behind the right leg. Slowly bend the right leg and move hips forward, keeping the left knee straight, the heels flat on the floor, and the back straight (Supplementary Figure S1). Hold this position for about 30 s and repeat with the other leg. |
| 2. Back of the thigh (hamstring) | While holding on to a chair, keep the left foot in front of the right foot with the knees straight and the heels on the ground. Bend the right knee, keep the heel on the ground and lean toward the chair (Supplementary Figure S2). Hold this stretch for about 30 s and repeat with switching legs. |
| 3. Front of the thigh (quadriceps) | Grab the top of the left foot behind while keeping the right leg straight and the knees close together. Hold on to the chair using the right hand for support and avoid leaning forwards or to the side (Supplementary Figure S3). Hold this position for about 30 s and repeat with the other leg. |
| 4. Inner thigh | Stand up and keep the feet greater than the hip distance apart with the toes pointing slightly outward. Shift the weight to the right leg and slowly lean towards the right side by bending the right knee, keeping the left leg straight. Rest hands on the thighs for support and bend forwards, pushing the hip backward (Supplementary Figure S4). Hold this position for 30 s and repeat with switching legs. |
| 5. Waist | Sit in a chair with the feet on the ground and seperated at a shoulder width. Lean forward with the elbows on the thighs and arms hanging towards the floor together (Supplementary Figure S5). Hold this position for about 30 s. |

| Stretching Site | Procedure |
| --- | --- |
| 6. Upper back | Sit in a chair with the feet on the ground.<br>Extend the arms forward, the hands together, the fingers interlaced, and the palms facing inward, so that the thumbs are up.<br>Contract the abdomen and gently drop the chin toward the chest (Supplementary Figure S6).<br>Hold this stretch for about 30 s. |
| 7. Neck | Bend the head slightly to the left. With the left hand, gently pull the head downward (Supplementary Figure S7).<br>Hold this stretch for about 30 s and repeat on the other side. |
| 8. Shoulder | Bring the left arm across the body and hold it with the right arm below the elbow (Supplementary Figure S8).<br>Hold this stretch for about 30 s, switch arms and repeat. |
| 9. Upper arm | Lift the left arm and bend it behind the head.<br>Place the right hand on the bent elbow (Supplementary Figure S9).<br>Hold this stretch for about 30 s and repeat with the right arm. |
| 10. Wrist | In the sitting position, extend the left arm and hold it in front of you, with the palm facing up.<br>Flex the left wrist, pointing the fingertips towards the floor.<br>Gently pull the fingers towards you using the right hand (Supplementary Figure S10).<br>Hold this stretch for about 30 s and repeat the stretch with the right hand. |

When explaining the stretching protocol before the intervention, we asked the participants, "Please do stretching every day," verbally and in writing. In addition, we provided them with a record sheet to record their blood pressure and the duration of time they performed the stretching. As such, we were able to assess the extent to which the participants could continue stretching voluntarily and avoid coercing participants.

*2.2. Measurement of Vascular Endothelial Function*

The primary endpoints were arteriosclerosis-related indicators and blood pressure, and the secondary endpoint was physical flexibility (measured value of long-sit anteflexion). We used three factors as the arteriosclerosis-related indicators in the present study: reactive hyperemia index (RHI), cardio ankle vascular index (CAVI), and ankle brachial pressure index (ABI).

RHI, which indicates the endothelial function, was measured by digital reactive hyperemic peripheral arterial pressure measurement (RH-PAT) using the Endo-PAT2000 device (Itamar Medical Ltd., Caesarea, Israel). This is a non-invasive technique that can use a pneumatic probe to record the pulse wave amplitude of the finger artery and capture a pulsatile plethysmograph. Prior to the measurement, it was confirmed that patients had fasted for at least four hours and refrained from caffeine, which can affect vascular tone, for at least eight hours. It was performed in a quiet and dim room at $25 \pm 1$ °C to reduce fluctuations in vascular tone [15]. After confirming rest on the bed, attach a dedicated probe for measuring fingertip pulse waves to the fingertips of both hands, measure the resting state for 5 min, then bleed for 5 min, and measure for 5 min after opening. RHI obtained using the EndoPAT2000 depends primarily on the ability of the vascular endothelium to produce nitric oxide and the reactivity of vascular smooth muscle. An RHI < 1.67 indicates vascular endothelial damage, 1.67 to 2.09 is considered normal, and 2.1 or higher is deemed to be good [16].

The CAVI and the ABI were measured with the Vasera VS-1500 Vascular Screening System (Fukuda Denshi, Tokyo, Japan) to record participants in the supine position. ABI was measured at the same time as CAVI. ECG electrodes were placed on both wrists, a microphone to detect heart sounds was attached to the sternum, and the cuffs were wrapped around both arms and ankles. After automatic measurement, the left and right mean values were used for CAVI and ABI analysis [17]. CAVI is a non-invasive arteriosclerosis index that is calculated based on the stiffness parameter β method measured by carotid echo, etc., and can easily and accurately measure the hardness peculiar to blood vessels that do not

depend on blood pressure [18]. CAVI < 8 is normal, $8.0 \leq CAVI < 9.0$ borderline, $9.0 \leq CAVI$ is suspected of arteriosclerosis [19]. ABI has been utilized in the management of peripheral arterial disease (PAD). The normal range of ABI is 1.00 to 1.40, and anomalous values are defined as 0.90 or less. ABI values between 0.91 and 0.99 are considered "borderline," and values above 1.40 indicate incompressible arteries [20].

### 2.3. Assessment of Stretching and Blood Pressure

For the evaluation of stretching, measurements were taken while bending forward. The forward bending was measured according to the method of the Japanese Ministry of Education, Culture, Sports, Science, and Technology. In brief, participants were asked to sit with their feet together, slowly bend forward and slide the box placed in front of them forward and as straight as possible to measure the distance the box moved. The measurement was taken twice, and the better value was recorded.

The exercise implementation rate was calculated by dividing the number of days the stretching was performed in a month by the number of days in that particular month.

During the experimental period, all participants self-measured their blood pressure at home twice daily, in the morning and the evening. Blood pressure was measured before breakfast in the morning and at a calm time after dinner. After the intervention was started, the average blood pressure was calculated for the measurement day $\pm$ three days of each month.

We recorded basic information (age, height, weight, drinking habits, exercise habits, and antihypertensive drug intake status). Participants with exercise habits were defined as "people who exercised for 30 min or more at a time, on two days or more a week, and during one year or more." Drinking habits were recorded based on the current drinking situation.

### 2.4. Statistical Analysis

Using CAVI as the primary endpoint, the required sample size, 34 participants, was calculated under the condition of a decrease in CAVI value due to stretch intervention as 0.3 with the standard deviation of 0.4, $\alpha$ error of 5%, and the power as 80%. The changes in arteriosclerosis-related indicators from baseline to six months were analyzed using the Friedman test, and the Wilcoxon signed-rank sum test analyzed the differences from six to seven months and seven to ten months. Repeated measures analysis of variance (ANOVA) was used for changes in blood pressure during the ten-month duration and changes in flexibility. Easy R (EZR) version 1.55 was used for statistical analysis, and the significance level was judged with a risk rate of less than 5%.

## 3. Results

### 3.1. Characteristics of the Participants

Although the required sample size was 34 participants, the COVID-19 pandemic has occurred worldwide, including in Japan, during the patient entry period. Therefore, it has been challenging to continue new participant enrollment. Finally, a total of ten participants were included and completed the protocol: three men and seven women. No patients had other chronic diseases such as diabetes except hypertension. The average age (mean $\pm$ SD) of the participants was $60.1 \pm 6.1$ years. The body weight (mean $\pm$ SD) of the participants was $61.4 \pm 6.4$ (kg), and the body mass index (BMI) (mean $\pm$ SD) was $24.5 \pm 2.6$ kg/m$^2$. Four participants (40%) had exercise habits, and six participants (60%) did not. Of the former, three participants walked regularly, and one participant practiced kendo (a form of Japanese martial arts). Five participants (50%) drank alcohol, and five participants (50%) did not (Table 2). The average period since a diagnosis of hypertension was 13.7 years. Among all the participants, nine participants took antihypertensive drugs, and one did not. Antihypertensive drugs used were angiotensin II receptor blocker alone in four cases, calcium channel blocker alone in one case, and combination drugs (both angiotensin II receptor blocker and calcium channel blocker) in four cases (Table 3).

**Table 2.** Characteristics of the participants (*n* = 10).

| Factors | Values |
|---|---|
| Age (mean ± SD) | 60.1 ± 6.1 (years) |
| Time since diagnosis of hypertension | 13.7 ± 9.6 (years) |
| Body weight (mean ± SD) | 61.4 ± 6.4 (kg) |
| Body mass index (mean ± SD) | 24.5 ± 2.6 (kg/m$^2$) |
| Exercise habit | |
|    yes | 4 (40%) |
|    no | 6 (60%) |
| Alcohol-drinking | |
|    yes | 5 (50%) |
|    no | 5 (50%) |
| Antihypertensive drugs | |
|    yes | 9 (90%) |
|    no | 1 (10%) |

SD, standard deviation.

**Table 3.** Types of antihypertensive drugs used.

| Antihypertensive Drug | Participants (*n* = 10) |
|---|---|
| Angiotensin II receptor blocker alone | 4 |
| Calcium channel blocker alone | 1 |
| Combination drug | 4 |
| No drug | 1 |

*3.2. Stretching Compliance*

The exercise implementation rate was used to evaluate the compliance of the stretching program implemented in the current study. The exercise implementation rate for the entire period, excluding the rest period, was almost 90%. The overall average was 87.9%, and the monthly average for eight participants was over 80% for all periods, and for two participants, it was close to 50% (Table 4).

**Table 4.** Exercise implementation rate (%).

| Patient | 1 mo. | 2 mo. | 3 mo. | 4 mo. | 5 mo. | 6 mo. | 8 mo. | 9 mo. | 10 mo. | Mean |
|---|---|---|---|---|---|---|---|---|---|---|
| 1 | 100 | 100 | 100 | 71.0 | 64.3 | 100 | 90.3 | 93.1 | 100 | 91.0 |
| 2 | 100 | 86.7 | 93.1 | 96.6 | 83.9 | 100 | 92.9 | 93.1 | 96.6 | 93.6 |
| 3 | 45.5 | 42.9 | 46.4 | 41.9 | 62.1 | 48.0 | 60.0 | 54.8 | 63.3 | 51.7 |
| 4 | 90.6 | 94.1 | 97.0 | 91.2 | 97.1 | 97.1 | 100 | 96.8 | 96.8 | 95.6 |
| 5 | 100 | 96.8 | 100 | 100 | 100 | 100 | 100 | 100 | 100 | 99.6 |
| 6 | 96.9 | 90.6 | 100 | 100 | 96.8 | 100 | 96.7 | 96.8 | 90.9 | 96.5 |
| 7 | 76.7 | 56.7 | 66.7 | 66.7 | 61.3 | 46.7 | 54.8 | 40.0 | 46.7 | 57.3 |
| 8 | 100 | 100 | 100 | 100 | 100 | 100 | 100 | 100 | 100 | 100 |
| 9 | 93.5 | 100 | 100 | 100 | 100 | 90.3 | 100 | 100 | 73.3 | 95.2 |
| 10 | 93.5 | 100 | 100 | 100 | 100 | 93.5 | 96.8 | 100 | 96.7 | 97.8 |
| Mean | 89.7 | 86.8 | 90.3 | 86.7 | 86.5 | 87.6 | 89.1 | 87.5 | 86.4 | 87.9 |

The changes in body flexibility associated with stretching were evaluated using anteflexion measurements. The anteflexion measurement before the intervention (mean ± SD) was 35.7 ± 13.9 cm. The average value was 43.9 ± 13.0 cm in the sixth month and 46.1 ± 14.7 cm in the seventh month, and the flexibility was maintained for one month even when the stretching was paused. In the tenth month, anteflexion measurement was 47.3 ± 14.3 cm, displaying a significant improvement compared to the pre-intervention value ($p < 0.001$; Table 5).

**Table 5.** Change in flexibility.

|  | Pre | 1 mo. | 3 mo. | 6 mo. | 7 mo. | 10 mo. | *p*-Value |
|---|---|---|---|---|---|---|---|
| Forward bending mean (SD) | 35.7 (13.9) | 40.1 (14.8) | 40.8 (3.8) | 43.9 (13.0) | 46.1 (14.7) | 47.3 (14.3) | <0.001 |

SD: standard deviation.

### 3.3. Changes in Blood Pressure

Blood pressure was measured twice daily at home to investigate the effect of stretching on the blood pressure. Nine out of 10 participants were on antihypertensive drugs and controlled their blood pressures within the normal range throughout this study. The systolic blood pressures at the start of the study were $125 \pm 10$ mmHg in the morning and $120 \pm 15$ mmHg in the evening, and the diastolic blood pressures were $79 \pm 9$ mmHg in the morning and $74 \pm 13$ mmHg in the evening. Nine out of 10 participants took an-tihypertensive drugs and controlled their blood pressures within the normal range. The systolic blood pressures in the tenth month were $125 \pm 10$ mmHg in the morning and $120 \pm 9$ mmHg in the evening, whereas the average diastolic blood pressures were on $78 \pm 8$ mmHg in the morning and $74 \pm 11$ mmHg in the evening. Overall, the blood pressure was well controlled throughout the study period (Figure 2).

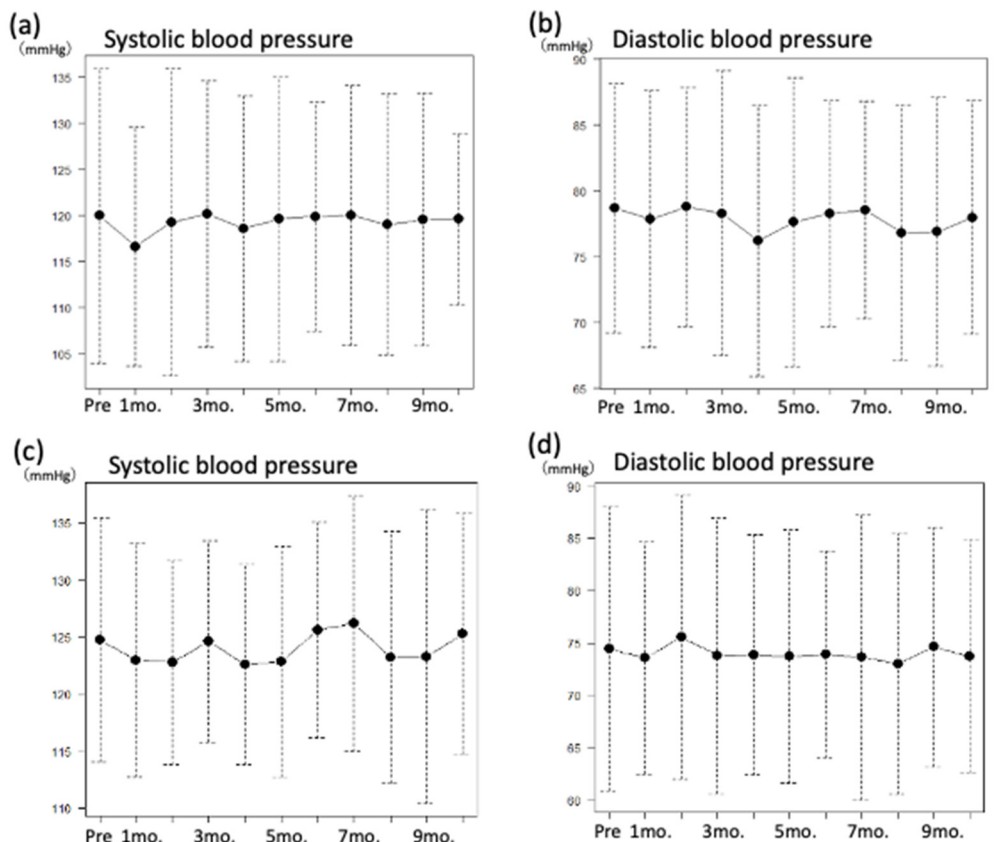

**Figure 2.** Change in blood pressure during the study period. (**a**) Change in systolic blood pressure in the morning, (**b**) change in diastolic blood pressure in the morning, (**c**) change in systolic blood pressure in the evening, (**d**) change in diastolic blood pressure in the evening.

### 3.4. Changes in Vascular Endothelial Function

In this study, the effect of stretch on vascular endothelial function was evaluated using three indicators: RHI, CAVI, and ABI.

Regarding changes in RHI, the baseline RHI value was 1.5, which is below the cut-off of 1.67 for vascular endothelial dysfunction. After introducing the stretching exercises, a slight improvement in RHI was observed up to the third month, which decreased in the

sixth month. However, no significant improvement was observed over the six months ($p = 0.063$; Table 6). During the resting phase (at the seventh month), the RHI value did not decrease (Table 7). When stretching was resumed from the seventh month, neither deterioration nor improvement of the RHI was observed (Table 8).

**Table 6.** Changes in arteriosclerosis index ($n = 10$).

|  | Pre | 1 mo. | 3 mo. | 6 mo. | *p*-Value |
|---|---|---|---|---|---|
| RHI | 1.50 (1.41,1.69) | 1.79 (1.52,2.15) | 1.77 (1.70,2.26) | 1.52 (1.42,1.97) | 0.063 |
| CAVI | 8.25 (6.05,9.60) | 7.95 (6.35,10.25) | 8.10 (5.90,9.20) | 7.80 (6.20,9.30) | 0.122 |
| ABI | 1.10 (0.97,1.25) | 1.11 (1.02,1.23) | 1.11 (1.06,1.17) | 1.08 (0.98,1.24) | 0.347 |

Data are given as median values (interquartile ranges).

**Table 7.** Changes in arteriosclerosis index at six and seven months ($n = 10$).

|  | 6 mo. | 7 mo. | *p*-Value |
|---|---|---|---|
| RHI | 1.52 (1.42,1.97) | 1.60 (1.35,2.08) | 1.000 |
| CAVI | 7.80 (6.20,9.30) | 7.88 (6.35,8.85) | 0.160 |
| ABI | 1.08 (0.98,1.24) | 1.16 (1.06,1.25) | 0.262 |

Data are given as median values (interquartile ranges).

**Table 8.** Changes in arteriosclerosis index at seven and ten months ($n = 10$).

|  | 7 mo. | 10 mo. | *p*-Value |
|---|---|---|---|
| RHI | 1.60 (1.35,2.08) | 1.68 (1.53,1.87) | 0.683 |
| CAVI | 7.88 (6.35,8.85) | 8.23 (6.35,10.75) | 0.192 |
| ABI | 1.16 (1.06,1.25) | 1.12 (1.06,1.18) | 0.306 |

Data are given as median values (interquartile ranges).

Regarding changes in CAVI, the CAVI value of 8.25 at the start of the intervention was in the boundary area, and it did not change significantly within the next six months (Table 6), during the rest period (Table 7), and after resumption (Table 8). Even though the CAVI value shifted from the boundary area to the normal area during the study period, no significant change was observed during the study period.

Regarding changes in ABI, the ABI value was within the normal range from the start to the 6th month (Table 6), during the rest period (Table 7), and after resumption (Table 8), and no significant change was observed during this study period.

## 4. Discussion

Management for all patients with suspected or diagnosed hypertension requires regular lifestyle advice, including diet and exercise [21]. Regular participation in aerobic exercise has been shown to lower office and ambulatory blood pressure of hypertensive individuals [22]. The optimal exercise modalities such as aerobic, resistance, and combined aerobic and resistance training have also been shown to improve arterial function in normotensive and hypertensive individuals [23,24]. A previous study reported that forced exercise (supervised, moderate-intensity aerobic exercise training for 40–50 min/session, 3 days/week for 12 weeks) followed by unsupervised exercise at home (at least 30 min/day, 1–2 days/week) as an efficacious tool to improve exercise adherence among a small sample of adults with hypertension [25]. However, it has been also suggested that the participation rates for hypertensive patients remain shockingly low and adherence to exercise training tends to decline following exercise programs [26]. Therefore, we introduced a less time-consuming and easy to perform low-intensity stretch protocol as a program that can be continued without being forced in the present study.

The main purpose of this study was to clarify whether the stretch exercise protocol we introduced could be easily implemented and continued by hypertensive patients. An eight-week stretching program has shown to be effective in lowering blood pressure among patients with high-normal blood pressure and stage I hypertension, indicating

the antihypertensive effect of the short-term stretching protocol [11]. Another study that used a more extended period of home-based exercise protocol (16 months) showed its effectiveness in improving functional capacity, blood metabolic profile, and blood pressure among hypertensive patients compared to a control group who did not comply with the exercise program [27]. However, 30 min moderate-intensity walking and stretching exercise program used in this trial seemed difficult to comply with and continue by middle-aged hypertensive patients with a busy life compared to the 10-min stretching protocol we adopted in the present study.

The intervention in this study was to perform the stretch protocol of about 10 min at any time of the day for 10 months, including a resting phase in the seventh month. Eight out of ten participants showed a 90% exercise rate each month. Most participants in this study had no exercise habits at the baseline, but when they decided to participate, they were willing to continue stretching. We believe that to be a major factor that influenced the high rates of exercise implementation. There were no dropouts or reported stretch injuries during the intervention period, and participants maintained a high exercise rate even after the rest phase. As such, we believe that the stretch protocol adopted in this study enhances participants' compliance because it was safe and easy to complete, regardless of the time or place defined. It was reported that helping patients identify emotionally rewarding, physically relevant activities, contingencies, and social support could increase exercise retention and promote desirable health outcomes [28].

The second purpose of this study was to investigate the effect of the stretching exercises we introduced on the blood pressure of hypertensive patients. However, we could not observe a significantly positive effect of the stretching intervention on the participants' blood pressure. This observation might be because the study participants had relatively controlled blood pressure. Nine patients took antihypertensive drugs in the morning, mainly angiotensin type II receptor blockers and Ca antagonists, and none of them took a large dose. Therefore, we believe that the presence or absence of antihypertensive drugs and their dose did not affect the blood pressure results of this stretching program. Although our study could not observe improved blood pressure by stretching interventions, previous studies have reported the effectiveness of 8 weeks of stretching interventions on the blood pressure of obese postmenopausal women [10] and healthy young men [29]. In addition, both aerobic and resistance exercise training has been reported to reduce blood pressure in the elderly with both prehypertension and hypertension [30].

In the present study, we observed a significant increase in flexibility from pre-intervention to 10 months, as shown in Table 5. The duration of each stretching in our study was around 30 s. A previous study has reported that a 30-s duration is an effective amount of time to sustain a hamstring muscle stretch to increase the range of motion [31]. Therefore, we believe that the significant improvement in flexibility we observed in this study is due to the sufficient time duration per motion. Our findings also revealed that flexibility could maintain even after 4-week resting of stretching exercises without deterioration and was compatible with a previous study which reported that a habitual stretch lasts at least four weeks after the effect is interrupted [32]. The increased flexibility during the study period also indicates that the participants had performed the stretching protocol correctly, demonstrating the high compliance and continuity levels of the intervention we used.

The present study evaluated the stretching effect on the arteriosclerosis index by measuring three indicators (RHI, CAVI, and ABI) during the study period. A recent systematic review based on 16 studies has reported that arterial stretching has a significant effect on reducing arterial stiffness [33]. However, we could not observe a positive effect in any of the indicators of the arteriosclerosis index. This could be due to the enrollment of participants with relatively controlled blood pressure. Because the arteriosclerosis indexes of our participants were already within the boundary to the normal area at the start of this study, significant improvement or change could not be detected.

The present study has several limitations. First, the number of participants was not sufficient. We initially calculated the required sample size as 34 people using CAVI

as the primary endpoint. However, while recruiting patients, the COVID-19 pandemic occurred worldwide, including in Japan; therefore, the enrollment of participants had to be terminated by 10 patients. The small sample size may have concealed a significant improvement in vascular endothelial function. Second, the intervention was applied to participants already taking oral antihypertensive treatment. Therefore, blood pressure and arteriosclerosis indicators were stable throughout the study period, and it is possible that the effects of stretching were underestimated. Third, confounding factors such as the dosage of anti-hypertensive medication, the intensity of other exercises performed, and stress may affect these results. However, we enrolled patients with relatively controlled blood pressure who were not on large doses of antihypertensive drugs. Therefore, it was assumed that the dose of the antihypertensive drug had no effect. In addition, we asked the patients whether they were doing other exercises or were stressed at the time of entry and each measurement timing. With regard to the other exercises, 3 people reported that they walked for 30 min twice a week. We assumed that it did not affect the result of this study as none of the patients complained of stress levels high enough to affect their blood pressure.

Finally, the stretching protocol adopted in this study significantly improved flexibility but did not affect arteriosclerosis indicators. However, the findings indicate that blood pressure and the arteriosclerosis index were well controlled during the intervention. In the future, it might be worthwhile to verify the effect of stretching on vascular endothelial function by increasing the sample size and including hypertensive patients who are not on antihypertensive medications. The stretching program adopted in this study can be completed in about 10 min and, therefore, can easily be implemented by hypertensive patients who are busy with work, etc. Although this study design includes a one-month rest period, stretching exercises over a period of 10 months showed good compliance, as indicated by the high implementation rate.

The stretching program used in this study was carried out by patients who attended the clinics. Since the short-time protocol could be easy to perform by hypertensive and pre-hypertensive patients, we think that this stretching program can be easily introduced to many clinic settings.

## 5. Conclusions

The compliance of the stretching protocol we used, evaluated by the exercise implementation rate for the entire period, was 90% or more; therefore, easy to perform and continue by hypertensive patients. However, we did not observe a significant positive effect of the stretching intervention on arteriosclerosis index or blood pressure in this study.

**Supplementary Materials:** The following supporting information can be downloaded at: https://www.mdpi.com/article/10.3390/clinpract12030036/s1, Figure S1: Stretching protocol.

**Author Contributions:** E.Y. and Y.K. (Yu Koyama) designed the study; E.Y., Y.K. (Yu Koyama), M.I. and K.M. conducted the study; E.Y., M.I. and K.M. collected the data; E.Y., Y.K. (Yuko Kikuchi) and K.O. analyzed the data; S.S., M.U. and Y.K. (Yu Koyama) summarized the data; E.Y., H.M.A. and Y.K. (Yu Koyama) wrote the manuscript; and Y.K. (Yu Koyama) had primary responsibility for the final content of the manuscript. All authors have read and agreed to the published version of the manuscript.

**Funding:** This research received no external funding.

**Institutional Review Board Statement:** This study was approved by the Niigata University Ethics Committee (approval ID: #2018-0025, the date of approval: 1 August 2018). This study was conducted in accordance with the principles of the Declaration of Helsinki.

**Informed Consent Statement:** Informed Consent was obtained from all patients before entering the study.

**Acknowledgments:** The authors would like to thank Masaomi Chinushi at the Division of Medical Technology, Niigata University Graduate School of Health Sciences for his excellent advice on designing this study.

**Conflicts of Interest:** The authors declare that there are no conflict of interest.

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
