# Peer review of "Compliance of Static Stretching and the Effect on Blood Pressure and Arteriosclerosis Index in Hypertensive Patients"

_clinpract, doi:10.3390/clinpract12030036_

Round 1

Reviewer 1 Report

I realize that authors have many journals to consider when they want to publish their work, so I appreciate your interest in Clinics and Practice;

The author presents an interesting study on compliance of static stretching and the effect on blood pressure and arteriosclerosis index in hypertensive patients. Overall, the study is well described and use adequate methods and analyses. Moreover, it presents good data on this very important topic. I believe that this is very important issue in field of Clinics and Practice section.

Minor concerns

Line 29 Abbreviation (ex: All CAVI, ABI, RHI etc.) have to show full name when first use. For example when first use term of CAVI, you have to describe full name (CAVI), and then second use term, you can only use abbreviation.

Line 31 “60.1±6.0” change to two decimals “60.1X±6.0X”

Line 33 p =0.00001, change to “p<0.001” (whole manuscript has to change to three decimals.

You should describe more detailed applications from this study in Discussion section.

You should add more backgrounds of this study in Introduction section.

You should add more arguments from this study’s result based on previous studies in Discussion section.

I recommend that this manuscript should be edited by an English professional editor for more readable. There are several grammatical errors.

Author Response

For Reviewer 1

Comments and Suggestions for Authors

  • Line 29 Abbreviation (ex: All CAVI, ABI, RHI etc.) have to show full name when first use. For example when first use term of CAVI, you have to describe full name (CAVI), and then second use term, you can only use abbreviation.

Answer

Thank you for your kind advice. We have corrected it according to the reviewer's point.

  • Line 31 “60.1±6.0” change to two decimals “60.1X±6.0X”

Answer

Thank you for your comment. We have corrected it in the abstract.

Along with this correction, we modified the numerical values in Table 2 and the corresponding text.

  • Line 33 p =0.00001, change to “p<0.001” (whole manuscript has to change to three decimals.

Answer

Thank you for your kind advice. We have corrected it according to the reviewer's point.

We noticed another similar point in Line 239, and corrected it accordingly.

  • You should describe more detailed applications from this study in Discussion section

Answer

Thank you for your kind advice. As per to the reviewer's advice, we have added a brief description in this regard at the end of the discussion.

  • You should add more backgrounds of this study in Introduction section.

Answer

Thank you for your kind advice. Following the reviewer's advice, we have modified the introduction section by adding more background. More reference papers were used, and the citations were revised accordingly.

  • You should add more arguments from this study’s result based on previous studies in Discussion section.

Answer

Thank you for your kind advice. What we would like to convey most about the results of this study was the good compliance of the stretching program we adopted. According to the reviewer's advice, we have described this point comparing with a stretching program used in a previous study in the Discussion section.

  • I recommend that this manuscript should be edited by an English professional editor for more readable. There are several grammatical errors.

Answer

The original version of our manuscript has already been edited by a professional English edition company, and the revised version of the manuscript was checked by a native speaker.

Reviewer 2 Report

This study support the idea of an alternative method for blood pressure and artherioslerosis index improvement in hypertensive patients. Even though, it is to be appreciated conducting a study on patients during pandemic conditions, the shortcomings limited this study to be performed only on a small number of participants, most with controlled blood pressure and exercise habits. Therefore, the results regarding the effect of static stretching on the investigated parameters were not quite relevant for the initial study. A larger number of participants, healthy individuals and hypertensive patients, with or without treatment or exercise habits, it could have made a difference. Moreover, tracking the effect of the exercises over a longer period of time is indicated due to well-known effects of long-term physical activity. I might also add that a follow-up study, after a period of time with ordinary life habits of the same participants, might be relevant to the initial study. However this study falls in the healthy living area promoting the importance of alternation of physical movement, even light exercises like static stretching, to drug therapy in hypertensive patients, leading also to improved flexibility at any age.

Author Response

For Reviewer 2

This study support the idea of an alternative method for blood pressure and artherioslerosis index improvement in hypertensive patients. Even though, it is to be appreciated conducting a study on patients during pandemic conditions, the shortcomings limited this study to be performed only on a small number of participants, most with controlled blood pressure and exercise habits. Therefore, the results regarding the effect of static stretching on the investigated parameters were not quite relevant for the initial study. A larger number of participants, healthy individuals and hypertensive patients, with or without treatment or exercise habits, it could have made a difference. Moreover, tracking the effect of the exercises over a longer period of time is indicated due to well-known effects of long-term physical activity. I might also add that a follow-up study, after a period of time with ordinary life habits of the same participants, might be relevant to the initial study. However this study falls in the healthy living area promoting the importance of alternation of physical movement, even light exercises like static stretching, to drug therapy in hypertensive patients, leading also to improved flexibility at any age.

Answer

Thank you for your very warm comments on our research. As the reviewer commented, the number of participants in this study was small. We hope to clarify the usefulness of this stretching program in the future by increasing the sample size and including hypertensive patients not currently on antihypertensive medications.

This manuscript is a resubmission of an earlier submission. The following is a list of the peer review reports and author responses from that submission.

Round 1

Reviewer 1 Report

  • The methodology is incomplete
  • Why those who quit smoking were excluded from the study?
  • More details about the stretching protocol are required. What was the duration and intensity of stretches?
  • Long sitting anterior flexion is not a valid method for the measurement of flexibility. More clarity is required on this issue
  • Was the BP measured before or after taking antihypertensive?
  • Number of participants seems to be too less for the study. Was the sample size calculated before the study?
  • There is a lot of cofounding factor which may affect your results such as the dosage of anti-hypertensive medication, the intensity of other exercise performed, stress.
  • Why there was no control group in the study?
  • The discussion is not in the standard format and is incomplete. The result of the study is not discussed properly
  • Most of the discussion especially about stretching is just the assumption of the author without any scientific evidence.

Author Response

For reviewer 1

Thank you for reviewing our manuscript, and we also thank you again for your appropriate comments. According to your comments, we have revised our manuscript, and also made slight change to the title of the manuscript.

Our reply to your comments are as follows:

Comments and Suggestions for Authors

  • The methodology is incomplete

Answer

Thank you for your important advice. We have added description on methodology more detail.

  • Why those who quit smoking were excluded from the study?

Answer

Thank you for your important comment. If smoker included, we have to evaluate several complicated conditions such as smoker or not, smoking history and the number of cigarettes per day, etc. We considered that the effects of smoking could cause variations in results, therefore, we excluded smokers in order to align the background as possible.

  • More details about the stretching protocol are required. What was the duration and intensity of stretches?

Answer

Thank you for your meaningful advice. We have added description of the stretching protocol more detail, as follows:

“The stretching program we used this time consists of 10 types of elements: calf, back of thigh, front of thigh, inside of thigh, waist, upper back, neck, shoulder, upper arm, wrist. One series took about 10 minutes and was a low intensity static stretch exercise that was not intense and could be easily performed during breaks at home or at work.”

  • Long sitting anterior flexion is not a valid method for the measurement of flexibility. More clarity is required on this issue

Answer

Thank you for your meaningful advice. We have added the description on the stretching protocol and the measurement of flexibility into Methods. In this study, we adopted anterior flexion as measurement of flexibilityaccording to the method of the Japanese Ministry of Education, Culture, Sports, Science and Technology.

  • Was the BP measured before or after taking antihypertensive?

Answer

Thank you for your meaningful comment. Blood pressure measurement in the morning was before taking antihypertensive drugs, and blood pressure measurement in the evening was after dinner and before bedtime. Nine people took antihypertensive drugs, and all of them had a schedule of taking antihypertensive drugs only in the morning, and there were no patients who took antihypertensive drugs immediately before the blood pressure measurement at night. Based on the above, we believe that the time taken for the antihypertensive drug did not affect the blood pressure measurement this time.

We have added following description: “Blood pressure was measured before breakfast, in the morning before taking, and at a calm time after dinner.”

  • Number of participants seems to be too less for the study. Was the sample size calculated before the study?

Answer

As the reviewer commented, the number of participants in this study was as small as 10 patients. We initially set the sample size to 34 people by using CAVI as the primary endpoint. However, while recruiting patients, COVID-19 pandemic has occurred all over the world including Japan, so the registration of participants had to be terminated by 10 patients.

We have added the description on sample size setting and also the reason of finally resulted small number at the Method and the Discussion.

  • There is a lot of cofounding factor which may affect your results such as the dosage of anti-hypertensive medication, the intensity of other exercise performed, stress.

Answer

This time, we entered patients with relatively controlled blood pressure, who did not have a large dose of antihypertensive drug, and it is considered that the dose of antihypertensive drug had no effect. In addition, we asked the patients whether they were doing other exercises or were stressed at the time of entry or at each measurement timing, but as for other exercises, 3 people walked for 30 minutes twice a week. I think that it did not affect the result of this time. None of the patients complained of stress enough to affect their blood pressure.

  • Why there was no control group in the study?

Answer

This time, in addition to the effect on arteriosclerosis factors such as CAVI, the purpose was to investigate the compliance of the stretch exercise introduced this time, and no control group was set. In the future, we wish to conduct a study comparing the effects of stretch on arteriosclerosis factors, if COVID-19 epidemic is over.

  • The discussion is not in the standard format and is incomplete. The result of the study is not discussed properly

Answer

We apologize for the non-standard format of our Discussion of the initial manuscript. This time, Discussion has been completely revised.

  • Most of the discussion especially about stretching is just the assumption of the author without any scientific evidence.

Answer

We think the Reviewer's comment is totally correct, and therefore, we have made a major rewrite of the Discussion.

Reviewer 2 Report

Thank you for your submitted manuscript entitled, “  Effect of Static Stretching on Arteriosclerosis Index and Blood 2 Pressure in Hypertensive Patients’’. The article is interesting and well written. However, I have a few comments that I suggest you consider before publishing the text.

ABSTRACT

  • Clarify the patients’ health background
  • Could be a relevant conclusion of the present study to find what is important to know.

INTRODUCTION

  • The introduction is consistent and easy to follow. Hypotheses are clearly formulated.
  • The Authors should clarify the actual heritage of this study. I am concerned about the originality of the present study.

METHOD

  • How was sample size determined? (Sampling technique!)

STATISTICAL ANALYSIS

  • Please, present methods of data analysis and criterion of results interpretation.
  • Please define specificity and sensitivity in this research context

RESULTS

  • Results description is a little chaotic and insufficient. Please, add some introductions to the description of the results and indicate what and why you did. Each result presented in the tables should be commented on in the text. Without that, readers do not know how to interpret the tables.

DISCUSSION

  • Discussion should be more based on the literature.

CONCLUSION

  • Why might one want to cite this paper? What is the true impact of the literature?

REFERENCES

  • Please adapt and improve based on the guidelines of the journal

Author Response

For reviewer 2

Thank you for reviewing our manuscript, and we also thank you again for your appropriate comments. According to your comments, we have revised our manuscript, and also made slight change to the title of the manuscript.

Our reply to your comments are as follows:

ABSTRACT

  • Clarify the patients’ health background

Answer

Thank you for your important comment. According to the reviewer's comment, we have added the data of body weight and BMI as patient background to the "Characteristics of the participants" and Table 1.

  • Could be a relevant conclusion of the present study to find what is important to know.

Answer

Thank you for your meaningful comment. In this study, it was also the purpose of the study to confirm the compliance of the stretching program we used, and good result on compliance was obtained, therefore we have described it in the conclusion of Abstract.

INTRODUCTION

  • The introduction is consistent and easy to follow. Hypotheses are clearly formulated.

Answer

Thank you for your kindly comment.

  • The Authors should clarify the actual heritage of this study. I am concerned about the originality of the present study.

Answer

Thank you for your kindly comments. According to your advice, we have added the description of the compliance of the stretching exercise we introduced this study, because the good compliance was the very significantly good result of our study.

METHOD

  • How was sample size determined? (Sampling technique!)

Answer

As the reviewer commented, the number of participants in this study was as small as 10 patients. We initially set the sample size to 34 people by using CAVI as the primary endpoint. However, while recruiting patients, COVID-19 pandemic has occurred all over the world including Japan, so the registration of participants had to be terminated by 10 patients.

We have added the description on sample size setting and also the reason of finally resulted small number at the Method and the Discussion.

STATISTICAL ANALYSIS 

  • Please, present methods of data analysis and criterion of results interpretation.

Answer

Thank you for your meaningful comment. As the reviewer commented, we have added the description of our data and criterion of each factor, especially, on RHI, CAVI and ABI at Method.

  • Please define specificity and sensitivity in this research context

Answer

In this research measurement, continuous variables are the main analysis targets, therefore, we did not examinespecificity and sensitivity.

RESULTS

  • Results description is a little chaotic and insufficient. Please, add some introductions to the description of the results and indicate what and why you did. Each result presented in the tables should be commented on in the text. Without that, readers do not know how to interpret the tables.

Answer

Thank you for your helpful comment. We have added explanations and/or introductions to the description of the results. And also, we have added the description corresponding to each table.

DISCUSSION 

  • Discussion should be more based on the literature.

Answer

We apologize for the non-standard format of our Discussion of the initial manuscript. This time, Discussion has been completely revised.

CONCLUSION

  • Why might one want to cite this paper? What is the true impact of the literature?

Answer

Thank you for your meaningful comment. In this study, it was also the purpose of the study to confirm the compliance of the stretching program we used, and good result on compliance was obtained, therefore we have described it as Conclusion.

REFERENCES

  • Please adapt and improve based on the guidelines of the journal

Answer

We apologize for our incomplete format of the description of references. We have corrected the description of references.

Reviewer 3 Report

The purpose of this study was to investigate the effect of static stretching on arteriosclerosis index and blood pressure in hypertensive patients.

In PubMed engine, we can find 30,094 articles using keyword “Effect of Static Stretching”. Moreover, there has too many articles regarding to static stretching and arteriosclerosis index and blood pressure. Namely, this topic has been well documented and reported in sport science field and there has many evidences and reported about static stretching on arteriosclerosis index and blood pressure. For this reason, this study has not scientific originality and novelty.

Moreover, the participants are serious small numbers (n=10) and no control group. Therefore, this manuscript is clumsy, and the result table and interpretation of the results are beginner level. For this reason, overall there is no rational of the importance of this study in sports and health science field.

Author Response

For reviewer 3

Thank you for reviewing our manuscript, and we also thank you again for your appropriate comments. According to your comments, we have revised our manuscript, and also made slight change to the title of the manuscript.

Our reply to your comments are as follows:

Your Comment

The purpose of this study was to investigate the effect of static stretching on arteriosclerosis index and blood pressure in hypertensive patients.

In PubMed engine, we can find 30,094 articles using keyword “Effect of Static Stretching”. Moreover, there has too many articles regarding to static stretching and arteriosclerosis index and blood pressure. Namely, this topic has been well documented and reported in sport science field and there has many evidences and reported about static stretching on arteriosclerosis index and blood pressure. For this reason, this study has not scientific originality and novelty.

Moreover, the participants are serious small numbers (n=10) and no control group. Therefore, this manuscript is clumsy, and the result table and interpretation of the results are beginner level. For this reason, overall there is no rational of the importance of this study in sports and health science field.

Answer

Thank you for your important comments.

We apologize and regret that our research content does not suit the reviewer's intentions. Certainly, the number of participants in this study was very small, and it might be difficult to make a sufficient comparison. We explain the reason for this to also other reviewers: the number of participants in this study was as small as 10 patients. We initially set the sample size to 34 people by using CAVI as the primary endpoint. However, while recruiting patients, COVID-19 pandemic has occurred all over the world including Japan, so the registration of participants had to be terminated by 10 patients.

Round 2

Reviewer 1 Report

The authors tried to improve the manuscript. However, some of the major issues are not being addressed 

  1. The stretching protocol is not presented properly. What about the duration and type of stretch? I suggest adding a table for the stretching protocol
  2. Lack of adequate sample size is a major concern
  3. There is a lot of confounding factor which may affect your results -further clarification is required on this?
  4. The discussion is still not in the standard format. 

Reviewer 2 Report

The manuscript was improved and can be accepted

Reviewer 3 Report

My opinion about the article remains the same of the first revision of manuscript. Most of the issues that I advanced to you cannot be repaired. A new study would be needed to make these things suitable. The clarifications provided do not solve the problem.